# Comparison of Volatile and Non-Volatile Compounds of Ice-Stored Large Yellow Croaker (*Larimichthys crocea*) Affected by Different Post-Harvest Handling Methods

**DOI:** 10.3390/foods14030431

**Published:** 2025-01-28

**Authors:** Yao Zheng, Yuan Ma, Na Lin, Xu Yang, Junjie Wu, Quanyou Guo

**Affiliations:** 1East China Sea Fisheries Research Institute, Chinese Academy of Fishery Sciences, Shanghai 200090, China; zhengyao@ecsf.ac.cn (Y.Z.); mayuan6842@163.com (Y.M.); linna@ecsf.ac.cn (N.L.); yangxu@ecsf.ac.cn (X.Y.); wujj@ecsf.ac.cn (J.W.); 2School of Health Science and Engineering, University of Shanghai for Science and Technology, Shanghai 200093, China

**Keywords:** cultured large yellow croaker, spinal cord cutting, bleeding, GC-IMS, metabolomics

## Abstract

To compare the impact of different post-harvest handling methods on volatile and non-volatile compounds, a total of 54 live large yellow croakers were subjected to commercial slaughter (CS), spinal cord cutting (SCC), or spinal cord cutting and bleeding (SCCB). The fish samples were ice-stored for 72 h, followed by the analysis of volatile compounds using gas chromatography–ion mobility spectrometry and non-volatile compounds using LC-MS-based untargeted metabolomics. The results revealed the detection of a total of 28 volatile organic compounds, with 23 being successfully identified, predominantly including alcohols, aldehydes, esters, ketones, and heterocyclic compounds. Substances such as (E)-2-nonenal and 2-butanone are highly sensitive to post-harvest handling methods during ice storage. Furthermore, 943 non-volatile metabolites were identified, showing significant differences in 180, 100, 117, and 186 metabolites across comparisons of SCC 0 h/CS 0 h, SCCB 0 h/CS 0 h, SCC 72 h/CS 72 h, and SCCB 72 h/CS 72 h, respectively. Notably, the altered metabolic pathways mainly involved fatty acid and amino acid metabolism, including pathways like glycerophospholipid metabolism and arginine biosynthesis. This study revealed the potential mechanisms underlying the enhancement of fish quality through spinal cord cutting and bleeding.

## 1. Introduction

Large yellow croaker (*Larimichthys crocea*) is a prominent marine-farmed fish in China, characterized by its distinctive golden color, delicious flavor, and abundance of high-quality proteins and polyunsaturated fatty acids [1]. With a production of 280,997 tons in 2023, it ranked second in terms of marine-farmed fish species in China [2]. After harvesting, these fish are predominantly distributed in the form of ice storage [3]. However, during this process, factors such as endogenous enzymes and microorganisms contribute to the inevitable quality deterioration, directly impacting consumer acceptance and commercial value. Various attempts have been made to degrade the quality deterioration of large yellow croaker during ice storage, including treatments with electrolytic water [4], ozonated slurry ice [5], and pulse light [6]. Currently, there is a growing focus on the process of post-harvest handling, which serves as a crucial link between aquaculture and distribution. Numerous studies confirmed that by utilizing appropriate post-harvest handling methods, both quality standards and animal welfare can be ensured [7,8,9].

The post-harvest handling of fish typically involves procedures such as slaughtering, bleeding, gutting, etc. Some scholars defined this stage as the Quality Determination Period (QDP), which serves as the cradle of quality and impacts storage stability during subsequent cold chain distribution [9]. It is well known that the stress experienced by fish during slaughter has a direct impact on the rigor mortis process and the quality of the fish. Common slaughter methods include air asphyxia, ice asphyxia, CO_2_ narcosis, electronarcosis, and spinal cord cutting. Acerete et al. [10] found that asphyxia causes a higher stress response and faster quality deterioration compared to CO_2_ narcosis in European sea bass (*Dicentrarchus labrax*). Rainbow trout (*Oncorhynchus mykiss*) slaughtered by electronarcosis experienced an immediate loss of consciousness and presented higher levels of pH and muscle glycogen compared to ice asphyxiation [11]. Spinal cord cutting is a quick slaughter procedure that immediately stops the struggling of the fish’s body by destroying the brain (spinal cord) with a sharp object [12]. This method is commonly employed in Japan for handling high-value fish species, ensuring both quality and animal welfare [13]. Regarding bleeding or blood removal, this process can improve flesh color and reduce unpleasant odors by lowering the oxidation levels of myoglobin and lipids in fish during storage. Olsen et al. [14] observed improved fillet whiteness of Atlantic cod (*Gadus morhua*). However, the molecular mechanisms underlying the quality differences remain unclear, especially regarding the alterations in small molecule metabolites.

Volatile metabolites or volatile organic compounds (VOCs) are low-molecular-weight compounds typically below 300 Da [15]. They are more sensitive to metabolic changes and exhibit faster alterations compared to macromolecules such as proteins. Meanwhile, VOCs can be perceived through the olfactory sensory organs in the nasal cavity, playing a crucial role in the odor characteristics of fish products [16]. During the cold storage of tilapia (*Oreochromis mossambicus*) fillets, VOCs initially dominated by hydrocarbons and aldehydes, evolving into a complex composition by the end of storage, including hydrocarbons, aldehydes, ketones, alcohols, phenols, esters, and other compounds [17]. Significant changes in octanal, 3-methylbutanal, and ethyl acetate were found in dry-cured fish (*Mylopharyngodon piceus*) at both 4 °C and 25 °C [18]. Gas chromatography-ion mobility spectrometry (GC-IMS) is an advanced methodology that offers advantages such as faster detection speed, higher sensitivity, and suitability for real-time monitoring of trace-level compounds [19]. Similarly, metabolomics is a powerful tool for qualitative and quantitative analysis of non-volatile compounds of low molecular weight, providing comprehensive metabolic information owing to its high-throughput characterization [20]. Metabolomics can effectively identify the profiling and transformation of global metabolic profiles, making it particularly valuable for sensitively reflecting metabolic response information influenced by both endogenous and exogenous factors [21]. Du et al. [22] employed metabolomics to investigate the flesh quality regulation of crucian carp (*Carassius auratus*) subjected to a short-term micro-flowing water system. The enhancement of flesh quality in crucian carp was attributed to the TCA cycle, ornithine cycle, purine metabolism, and amino acid catabolism. The metabolic profiles of bottom cultured scallops (*Mizuhopecten yessoensis*) subjected to mechanical shock were investigated through metabolomics [23]. The authors observed that the biosynthesis of amino acids, particularly cysteine and methionine metabolism, was disrupted by mechanical shock. Zhang et al. [24] conducted a review of the stress responses affecting fish muscle quality within the industrial chain framework. They highlighted the potential of metabolomics in exploring the mechanisms by which stress response impact fish muscle quality.

In our previous research, it was observed that the treatment of spinal cord cutting reduced the rate of quality deterioration in pufferfish during frozen storage, as evidenced by thawing loss, cooking loss, and springiness [7]. Building upon this, this study further enhanced the treatment of spinal cord cutting by incorporating bleeding in the case of large yellow croaker, and the molecular mechanisms that underlie the quality changes were explored. Changes in volatile and non-volatile metabolites were scrutinized using GC-IMS- and LC-MS-based untargeted metabolomics, respectively. The objectives of this study aimed to elucidate the impact of post-harvest handling methods on these low-molecular-weight metabolites and to offer a theoretical basis for mitigating the deterioration of quality through appropriate post-harvest handling techniques.

## 2. Materials and Methods

### 2.1. Sample Preparation

In June 2024, live cultured large yellow croaker (weight: 392.3 ± 21.8 g, length: 34.7 ± 1.4 cm, n = 60) were purchased from Xuanshan Marine Ranching Co. located in Zhoushan, Zhejiang, China. All the fish were from the same batch, caught from the same pond, ensuring consistency in factors such as age, diet, and aquaculture conditions. Each fish was placed individually in polyethylene bags filled with seawater and oxygen and transported to our laboratory in Shanghai within five hours. Upon arrival, the live fish were immediately transferred to pre-prepared tanks equipped with seawater and oxygen pumps. After a 30 min acclimation period, the live cultured large yellow croaker was used for subsequent experiments.

A total of 54 large yellow croaker with similar sizes and good vitality were randomly divided into three groups of the same size: (1) the commercial slaughter group (CS), in which the live fish were slaughtered by direct immersion into a foam box filled with a mixture of ice and water (3:1 ratio) to induce asphyxiation for 30 min; (2) the spinal cord cutting group (SCC), in which the live fish were quickly slaughtered by cutting the spinal cord at the head region within 5 s; (3) the spinal cord cutting and bleeding group (SCCB), in which the live fish were slaughtered by cutting the spinal cord at the head region, followed by cutting the gill and placing the fish in room temperature water for 10 min to allow for bleeding. After completing the aforementioned procedures, samples from all three groups were placed in foam boxes filled with crushed ice for storage (Figure 1). At 0, 6, 12, 24, 48, and 72 h, dorsal muscle was collected and immediately frozen using liquid nitrogen for VOC analysis. Samples at 0 and 72 h were also subjected to metabolomics analysis.

### 2.2. Reagents

Methanol, acetonitrile, formic acid, and isopropanol are all chromatography-grade chemicals (Thermo Fisher Scientific, Waltham, MA, USA).

### 2.3. Gas Chromatography–Ion Mobility Spectrometry (GC-IMS)

Volatile organic compounds were analyzed through GC-IMS (FlavourSpec^®^, Gesellschaft für Analytische Sensorsysteme mbH, Dortmund, Germany) following a previously described method with minor modifications [25]. A quantity of 2.0 ± 0.1 g of the dorsal muscle sample was accurately weighed and incubated at 60 °C for 15 min at a rotation speed of 500 rpm in a 20 mL headspace vial. The sample was injected using an automated headspace injection method with an injection volume of 500 μL and an injection needle temperature of 65 °C. A strong polar chromatographic column (MXT-WAX, 30 m × 0.53 mm) was used, with a column temperature set at 60 °C. Nitrogen gas (≥99.999%) served as the carrier gas, following a flow rate program: an initial flow rate of 2 mL/min held for 2 min, increased to 8 to 10 mL/min over 8 min, then to 100 mL/min over 10 min, and held for 10 min. The total analysis time was 30 min, with the IMS detector maintained at 45 °C. Volatile compounds were identified according to retention index and drift time using the built-in NIST database and IMS database in the software.

### 2.4. Metabolomic Analysis

A precise weight of 50 mg of the fish muscle sample was taken and subjected to metabolite extraction using a solution of 400 μL methanol–water (4:1, *v*/*v*) containing 0.02 mg/mL L-2-chlorophenylalanine. The mixture was allowed to settle at −10 °C and then processed using a high-throughput tissue crusher (Wonbio-96c, Shanghai Wanbo Biotechnology Co., LTD, Shanghai, China) at 50 Hz for 6 min, followed by ultrasound treatment at 40 kHz for 30 min at 5 °C. Subsequently, the samples were incubated at −20 °C for 30 min to precipitate proteins. After centrifugation at 13,000× *g* for 15 min at 4 °C (5430R, Eppendorf, Hamburg, Germany), the resulting supernatant was carefully transferred to sample vials for subsequent LC-MS analysis. Additionally, 20 μL of supernatant was taken from each sample and mixed together as a quality control (QC) sample to assess the stability of the entire detection process.

Metabolites were analyzed using an ultra-high performance liquid chromatography (Acquity, Waters, MA, USA) tandem time-of-flight mass spectrometry system (Triple TOF 6600, AB SCIEX, Tokyo, Japan) according to the previous method [26]. The chromatographic conditions involved the separation of a 10 μL sample using an HSS T3 column (100 mm × 2.1 mm i.d., 1.8 μm). The mobile phases consisted of 0.1% formic acid in water–acetonitrile (95:5, *v*/*v*) (solvent A) and 0.1% formic acid in acetonitrile–isopropanol–water (47.5:47.5, *v*/*v*) (solvent B). The solvent gradient program was as follows: 0% B was maintained from 0 to 0.5 min, followed by a linear increase from 0 to 25% B from 0.5 to 2.5 min, a linear increase from 25 to 100% B from 2.5 to 9 min, a hold at 100% B from 9 to 13 min, a linear decrease from 100 to 0% B from 13 to 13.1 min, and finally a hold at 0% B from 13.1 to 16 min for system equilibration. The sample injection volume was 10 μL, the flow rate was set to 0.4 mL/min, and the column temperature was maintained at 40 °C. The chromatography system was coupled to a quadrupole-time-of-flight mass spectrometer equipped with an electrospray ionization (ESI) source, which operated in both positive and negative ionization modes. The optimal conditions were set as follows: the source temperature was maintained at 550 °C, the curtain gas (CUR) pressure was set to 30 psi, and both Ion Source Gas1 and Gas2 pressures were set to 50 psi. The ion-spray voltage floating (ISVF) was set at −4000 V for negative mode and 5000 V for positive mode. The declustering potential was set to 80 V, and the collision energy (CE) was set to vary from 20 to 60 eV for MS/MS using a rolling technique. Data acquisition was performed using the information-dependent acquisition (IDA) mode, and the detection range was set from 50 to 1000 *m*/*z*.

### 2.5. Data Analysis

The experimental data underwent one-way ANOVA analysis using SPSS (SPSS 22.0, SPSS Inc., Chicago, IL, USA) to calculate the variance. Tukey’s Test was employed to identify significant differences at the 5% level. The metabolomic data were searched and identified using HMDB (http://www.hmdb.ca/), Metlin (https://metlin.scripps.edu/), and Majorbio Database. Enrichment bubble plotting and metabolic pathway elucidation were achieved with the pathway database of the Kyoto Encyclopedia of Genes and Genomes (KEGG, https://www.genome.jp/kegg).

## 3. Results and Discussion

### 3.1. Changes in Volatile Organic Compounds

The variation in VOCs in the dorsal muscle of large yellow croaker subjected to different post-harvest handling methods was analyzed by GC-IMS. The retention times and ion migration times are listed together with the compound name, CAS number, molecular formula, molecular weight (MW), reserved index (RI), retention time (RT), drift time (DT), and response peaks, as presented in Appendix A. Due to a large amount of data, VOCs were visualized using fingerprint spectra, where each row represents a sample, and each column denotes a specific compound (Figure 2). A total of 28 signal peaks were detected, and 23 compounds were identified based on the GC-IMS database. The five unidentified compounds were labeled as V1, V2, V3, V4, and V5. The identified 23 compounds include 4 dimers, and after merging identical compounds, the 19 VOCs consist of 7 alcohols, 4 aldehydes, 3 esters, 3 ketones, and 2 heterocyclic compounds. Hydrocarbons such as alkanes and alkenes commonly found in meat were not detected because their proton affinities are lower than that of water, rendering them non-ionizable. Additionally, hydrocarbons have high odor thresholds and are usually not considered as key aroma components [27]. Similarly, Yang et al. [28] identified 27 VOCs in large yellow croaker, including compounds such as 1-propanol, 1-butanol, 1-penten-3-ol, 2-methylbutanal, ethyl acetate, and 2-methylbutanal, which were also confirmed in our study. Zhao et al. [29] also reported that 26 VOCs were identified in large yellow croaker during frozen storage, with the most abundant types being aldehydes, alcohols, and ketones, which aligns with the results of our study.

It can be seen that there were obvious differences in the VOCs as affected by different post-harvest handling methods. At the beginning of ice storage (0 h), the levels of (E)-2-nonenal, p-cymene, and V5 of the CS group were lower than those in the SCC and SCCB groups. Conversely, the contents of butanol, methyl 2-methylbutanoate, V3, and V4 in the CS group were higher than those in the SCC and SCCB groups. It is well known that the formation of VOCs in meat primarily arises from enzymatic reactions, lipid oxidation, and microbial metabolism [30]. In the CS group, muscle movement persists until the autonomic nerve eventually ceases its function, resulting in additional energy consumption and metabolic changes [12]. Therefore, the enzymatic reactions and lipid oxidation may have been influenced by the treatment of spinal cord cutting, thereby altering the formation of VOCs. While for the SCC and SCCB groups, the effects of bleeding treatment may not have manifested yet at this point, resulting in a similarity in VOCs profile. As the storage time prolonged, the levels of (E)-2-nonenal and V5 in the CS group still remain lower than those in the SCC and SCCB groups. Starting from 24 h, the levels of 2-butanone and propanol in the SCCB group were consistently lower than those in the SCC group. Generally, 2-butanone is more likely to be generated from lipid oxidation [31], which can be accelerated by pro-oxidative substances in the blood. The SCCB group underwent a bleeding process that removed a significant portion of the blood, including hemoglobin, which is a crucial pro-oxidant substance. Therefore, the lower level of 2-butanone in the SCCB group may be attributed to the bleeding process that reduced the oxidation reactions in the muscle tissue during the ice storage. Similar results were also found in sea bass (*Lateolabrax japonicus*) that the levels of propanal were lower in bloodletting to death group compared to the ice faint to death group and the head shot control stun death group [32]. In brief, VOCs, as small molecular substances, are highly sensitive to metabolic differences induced by different post-harvest handling methods.

### 3.2. Identification of Metabolites

As for non-volatile compounds, a total of 943 metabolites were detected in cultured large yellow croaker subjected to different slaughter methods. According to the Human Metabolome Database (HMDB), the most comprehensive organism-specific metabolic database, the identified metabolites were primarily composed of lipids and lipid-like molecules, accounting for 46.07% of the total. Organic acids and derivatives accounted for 13.90%, organic oxygen compounds accounted for 6.69%, benzenoids accounted for 5.92%, organoheterocyclic compounds accounted for 5.92%, and nucleosides, nucleotides, and analogues accounted for 4.63% (Figure 3A). In our previous study on the muscle of obscure pufferfish (*Takifugu obscurus*), a total of 654 metabolites were identified, with lipids also comprising the largest proportion [7]. Similarly, a recent study on the depuration process of Pacific oyster (*Crassostrea gigas*) found that the most abundant metabolite categories among the 610 detected metabolites were lipids, organic acids, and nucleic acids [33].

### 3.3. Screening of Differential Metabolites

Given the large volume of metabolomics data, it is common to employ a combination of univariate and multivariate statistical analysis to screen differential metabolites. In this study, the differential metabolites were screened out based on the following criteria: (1) metabolites that exhibited a fold change of 1 or higher between two groups, with a P value of less than 0.05; and (2) metabolites with variable importance in projection (VIP) values of one or higher, as calculated through the Orthogonal Partial Least Squares Discriminant Analysis (OPLS-DA) model [34]. The predictive capability of the OPLS-DA model was assessed by calculating the Q2 value, with Q2 > 0.5 indicating a good predictive ability. The Q2 values for SCC 0 h/CS 0 h, SCCB 0 h/CS 0 h, SCC 72 h/CS 72 h, and SCCB 72 h/CS 72 h were 0.609, 0.382, 0.748, and 0.659, respectively, indicating a stable and reliable classification model was obtained.

The volcano plots of differential metabolites are shown separately in Figure 3B (SCC 0 h/CS 0 h), Figure 3C (SCCB 0 h/CS 0 h), Figure 3D (SCC 72 h/CS 72 h), and Figure 3E (SCCB 72 h/CS 72 h). The horizontal axis represents the fold change in metabolites between the two groups, while the vertical axis represents the statistical significance difference in metabolites. Each plot represents a specific metabolite, with point size indicating the VIP value. Red plots indicate significantly upregulated metabolites, green plots represent significantly downregulated metabolites, and blue plots represent differentially non-significant metabolites. At the beginning of ice storage, compared to the CS 0 h group, the SCC 0 h group had a total of 180 differential metabolites, with 52 upregulated and 128 downregulated. The SCCB group had 100 differential metabolites compared to the CS group, with 40 upregulated and 60 downregulated. At the end of ice storage, compared to the CS 72 h group, the SCC 72 h group had a total of 117 differential metabolites, with 45 upregulated and 72 downregulated. The SCCB group had 186 differential metabolites compared to the CS group, with 142 upregulated and 44 downregulated. Detailed information on the differential metabolites is presented in Appendix A. Similarly, 191 metabolites were identified as differential metabolites during the depuration (0 h, 24 h, and 48 h) of Pacific oyster (*Crassostrea gigas*) [33]. Since the storage time is similar to that of our study, and the samples have not undergone intense processing, with the quantity of differential metabolites comparable to our study. The above results indicated that the treatments of spinal cord cutting and bleeding significantly influence the metabolic status during the ice storage of large yellow croaker.

### 3.4. Differential Metabolites and Metabolic Pathway

#### 3.4.1. Differential Metabolites

The complete list of differential metabolites can be found in Appendix A, while the top 30 differential metabolites based on VIP values for each comparison group are displayed in Figure 4. Each column on the left corresponds to a sample, each row to a metabolite, with colors denoting the relative expression levels. On the right side are the VIP values of the metabolites, where the length of the bars signifies the contribution of VIP values, and the color indicates the significance of the differences. As shown in Figure 4A, the differential metabolites of SCC 0 h/CS 0 h mainly comprised indole-3-lactic acid, Gdp-glucose, 17-octadecynoic acid, etc. The differential metabolites of SCCB 0 h/CS 0 h mainly comprised aspartic acid, cystathionine, docosapentaenoic acid, etc. (Figure 4B). The differential metabolites of SCC 72 h/CS 72 h mainly comprised palmitoleic acid ethyl ester, glucosylsphingosine, avocadene acetate, etc. (Figure 4C). The differential metabolites of SCCB 72 h/CS 72 h mainly comprised arachidonic acid, L-lysine, L-ornithine, etc. (Figure 4D).

#### 3.4.2. Differential Metabolic Pathway

Using the screened differential metabolites, pathway enrichment analysis was conducted with the KEGG database to comprehensively understand the metabolic state of large yellow croaker under different post-harvest handling methods. At the beginning of storage (0 h), compared to the CS group, the SCC group exhibited changes in pathways such as cortisol synthesis and secretion, glycerophospholipid metabolism, linoleic acid metabolism, fat digestion and absorption, and others (Figure 5A). Lipids play diverse roles in biological metabolism, involving participation in energy metabolism and serving as precursors to volatile organic compounds (VOCs). The synthesis and secretion of cortisol have been shown to be closely related to stress responses in fish. Stress is transmitted through electrical signals to the brain, leading to the release of hormones and electrical impulses, which activate the hypothalamic–pituitary–interrenal system, ultimately resulting in the production of cortisol [24]. In this study, the treatment of spinal cord cutting in the SCC group instantaneously disrupted the autonomic nerve, markedly reducing the stress during the slaughter process. Consequently, there were alterations in the synthesis and secretion of cortisol. Glycerophospholipids, the most prevalent type of phospholipids, play a crucial role in flavor compound formation through their diverse compositions and structures. Zhang et al. [25] reported a significant association between phosphatidylcholine and phosphatidylethanolamine with flavor substances, suggesting their role as key precursors for flavor development. Glycerophospholipids could generate VOCs through both non-enzymatic oxidation and enzymatic oxidation pathways, thereby potentially impacting the flavor of food. A study on Beijing-You Chicken revealed that glycerophospholipids contribute to the flavor of breast meat, suggesting their role as key precursors for flavor development [35]. Apart from glycerophospholipid metabolism, the metabolism of linoleic acid, as an important unsaturated fatty acid, also underwent changes. Similar results were also found in pufferfish with different post-harvest handling methods [7] and in scallops (*Mizuhopecten yessoensis*) under hypoxic stress [9]. This indicates that the metabolism of linoleic acid is highly susceptible to stress factors and holds potential as a biomarker. The disturbance of linoleic acid metabolism may be attributed to the instability of its double bonds, which are prone to oxidation reactions under stress conditions. It is widely acknowledged that the oxidation reaction is mainly triggered by reactive oxygen species (ROS). Under the stress induced by slaughter, more ROS could be produced by the mitochondrial oxidation process, which cortisol encourages [24].

While for SCCB and CS group, glycerophospholipid metabolism and linoleic acid metabolism were still being identified at the beginning of storage (0 h). Additionally, changes in pathways such as arginine biosynthesis, alanine, aspartate, and glutamate metabolism; cysteine and methionine metabolism; and valine, leucine, and isoleucine metabolism were found (Figure 5B). Amino acids play vital roles as sources of energy, building blocks for protein synthesis, regulators of osmotic balance, and contributors to immune responses. In this study, differences in amino acid metabolism may primarily stem from energy metabolism. This is because the treatment of spinal cord cutting prevents muscle movement after death, whereas muscle movement is associated with faster energy consumption [13]. Arginine plays a vital role in the urea cycle, where it undergoes catalysis by arginase to generate urea and ornithine. A recent study reported that arginine can participate in an energy-conserving reaction in amino acid metabolism catalyzed by arginine synthetase [36]. Alanine, aspartate, and glutamate are derived from intermediates of central metabolism, primarily the citric acid cycle, which is a critical pathway for energy production [37]. While for cysteine and methionine metabolism, cysteine can be metabolized through multiple pathways to participate in energy metabolism. Isoleucine can participate in energy metabolism through the pentose phosphate pathway. Similar changes in amino acid metabolism were reported in turbot (*Scophthalmus maximus*) and salmon (*Oncorhynchus masou*), where amino acid content was affected by the treatment of spinal cord cutting and bleeding [8,13]. However, minor effects were noted in the levels of free amino acids in Atlantic salmon (*Salmo salar* L.) during ice storage [38]. This could be due to the differences in metabolic impacts resulting from the treatments employed in the study, which included crowding and fatigue. It is also worth noting that amino acid metabolism could influence the eating quality, with free amino acids recognized for their substantial contribution to taste sensation.

At the end of the storage (72 h), the differential metabolic pathways still primarily involved fatty acid metabolism and amino acid metabolism. Pathways of glycerophospholipid metabolism and biosynthesis of unsaturated fatty acids were identified in both SCC/CS and SCCB/CS (Figure 5C,D). Additionally, histidine metabolism and glycine, serine, and glutamate metabolism were also identified. Histidine, classified as a glucogenic amino acid, plays a role in energy production through degradation to pyruvate, α-ketoglutarate, and succinyl-CoA. It was reported that pre-mortem stress altered the histidine contents in Atlantic salmon (*Salmo salar* L.) during ice storage, and the authors speculated that histidine could inhibit the activity of cathepsin B and L at a specific concentration [38]. Glycine is a key substance in modulating the antioxidant system and also serves as an umami and sweet substance. In other words, the aforementioned results provide a theoretical basis from a small molecule perspective for the practical application of spinal cord cutting and bleeding post-harvest treatment in production. By employing appropriate post-harvest practices, quality can be enhanced while safeguarding animal welfare. The commercial benefits derived from improved quality can incentivize enterprises to adopt suitable post-harvest treatment methods.

## 4. Conclusions

In this study, a thorough analysis of volatile and non-volatile metabolites was conducted on large yellow croaker subjected to different post-harvest handling methods using GC-IMS and metabolomics techniques. A total of 28 compounds were detected, with 23 volatile organic compounds identified, predominantly comprising alcohols, aldehydes, esters, ketones, and heterocyclic compounds. Notably, compounds such as (E)-2-nonenal and 2-butanone were sensitive to the treatment of spinal cord cutting and bleeding. Regarding the non-volatile metabolites, a total of 943 metabolites were detected, with 180, 100, 117, and 186 significantly different metabolites in comparisons between SCC 0 h/CS 0 h, SCCB 0 h/CS 0 h, SCC 72 h/CS 72 h, and SCCB 72 h/CS 72 h, respectively. The differential metabolic pathways primarily involved in fatty acid metabolism and amino acid metabolism, such as glycerophospholipid metabolism and arginine biosynthesis. This could be attributed to the treatment of spinal cord cutting and bleeding, which alters energy metabolism and oxidative reactions. These findings contribute to enabling the seafood industry to enhance post-harvest handling methods, thereby improving both quality and animal welfare. Future research could concentrate on energy metabolism and delve deeply into the relationship between post-harvest handling methods and eating quality from the perspective of protein phosphorylation and other related aspects. It is essential to investigate whether improving post-harvest handling methods can reduce inputs during storage processes (such as temperature, packaging, additives, etc.), thereby achieving a reduction in quality loss from the source.

## Figures and Tables

**Figure 1 foods-14-00431-f001:**
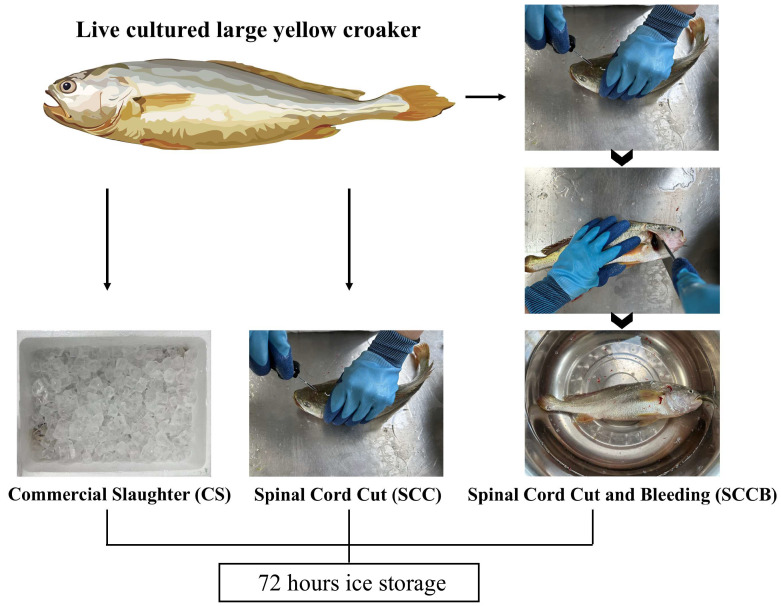
The experimental design and post-harvest handling methods in this study.

**Figure 2 foods-14-00431-f002:**
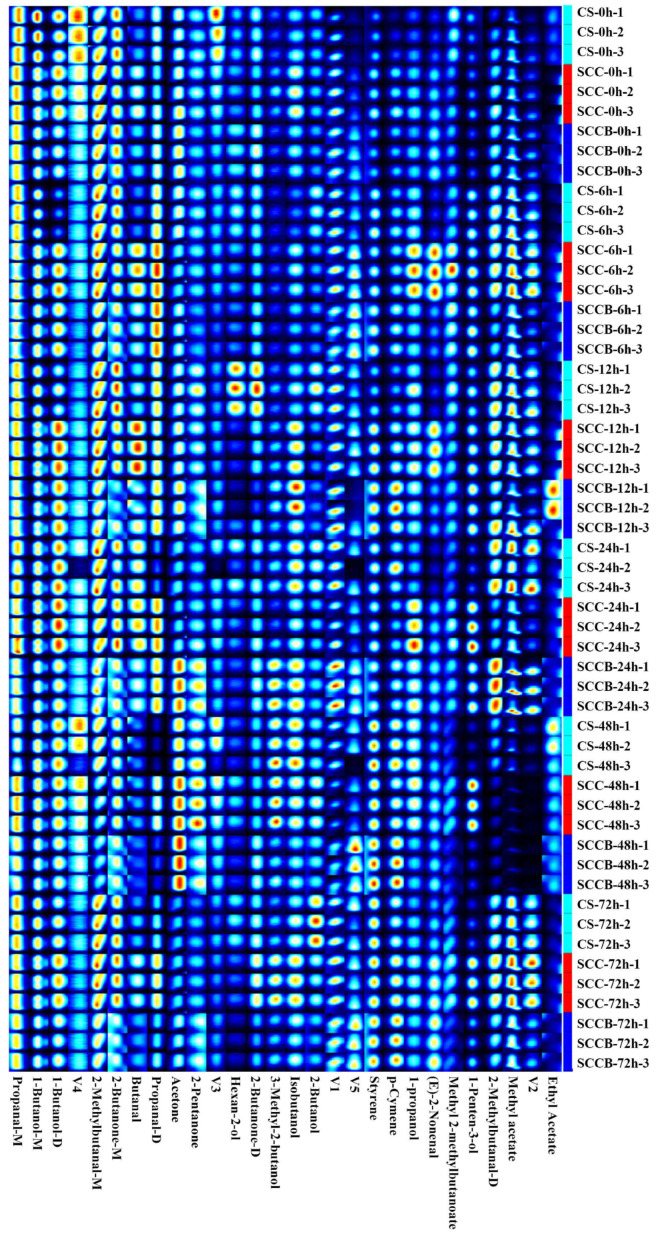
Fingerprint spectra of volatile organic compounds. Note: CS, the commercial slaughter group; SCC, the spinal cord cutting group; SCCB, the spinal cord cutting and bleeding group; V1, V2, V3, V4, and V5 represent five unidentified substances.

**Figure 3 foods-14-00431-f003:**
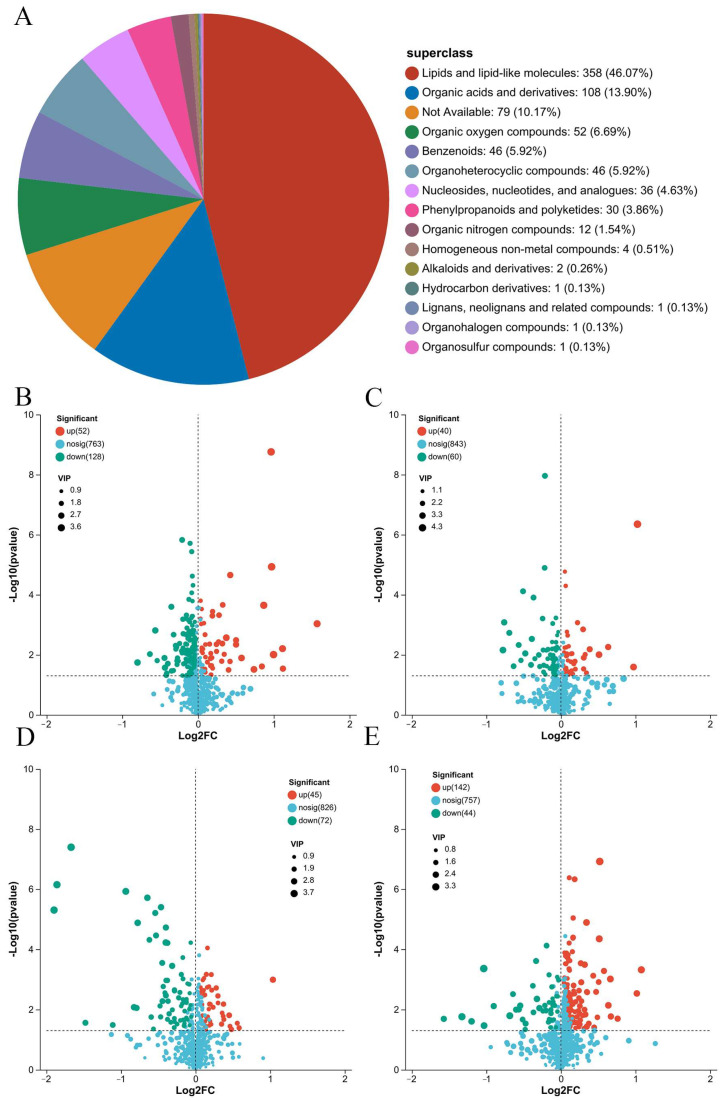
The identified metabolites and their superclass (**A**), and volcano plots of differential metabolites ((**B**) SCC 0 h/CS 0 h; (**C**) SCCB 0 h/CS 0 h; (**D**) SCC 72 h/CS 72 h; (**E**) SCCB 72 h/CS 72 h). Note: CS, the commercial slaughter group; SCC, the spinal cord cutting group; SCCB, the spinal cord cutting and bleeding group.

**Figure 4 foods-14-00431-f004:**
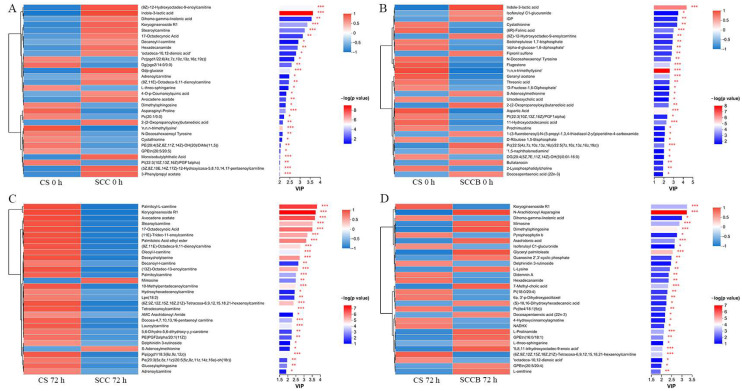
The VIP and *p*-value of differential metabolites, (**A**) SCC 0 h/CS 0 h; (**B**) SCCB 0 h/CS 0 h; (**C**) SCC 72 h/CS 72 h; (**D**) SCCB 72 h/CS 72 h. Note: CS, the commercial slaughter group; SCC, the spinal cord cutting group; SCCB, the spinal cord cutting and bleeding group; * represents *p* < 0.05, ** represents *p* < 0.01, *** represents *p* < 0.001.

**Figure 5 foods-14-00431-f005:**
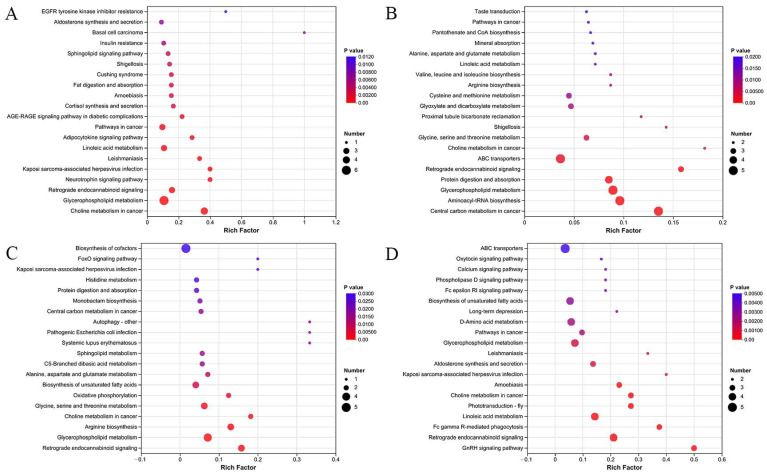
Pathway enrichment of differential metabolites ((**A**) SCC 0 h/CS 0 h; (**B**) SCCB 0 h/CS 0 h; (**C**) SCC 72 h/CS 72 h; (**D**) SCCB 72 h/CS 72 h). Note: CS, the commercial slaughter group; SCC, the spinal cord cutting group; SCCB, the spinal cord cutting and bleeding group.

## Data Availability

The original contributions presented in this study are included in the article and Appendix A. Further inquiries can be directed to the corresponding author.

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
