# Peer review of "Comparison of Volatile and Non-Volatile Compounds of Ice-Stored Large Yellow Croaker (Larimichthys crocea) Affected by Different Post-Harvest Handling Methods"

_foods, 2025, doi:10.3390/foods14030431_

Round 1
Reviewer 1 Report
Comments and Suggestions for Authors
This study provides novel insights into how post-harvest handling techniques influence metabolic pathways, which in turn affect the quality and sensory characteristics of large yellow croaker during storage. The findings are highly relevant for the seafood industry, addressing both quality control and animal welfare concerns. Comments and Suggestions for Improvement:
Abstract (Lines 14–30): Please include specific quantitative findings in the abstract, such as the number of metabolites altered by each handling method and the key volatile compounds identified.
Introduction (Lines 31–97): Please clarify how this study builds on previous research, particularly on post-harvest handling of large yellow croaker. Highlight the novelty of combining GC-IMS and metabolomics to explore these impacts.
Methods - Sample Preparation (Lines 98–117): Please provide more details on how fish were selected for uniformity (e.g., age, diet, or aquaculture conditions).
Methods - GC-IMS and LC-MS Analysis (Lines 124–169): Please explain why specific parameters for GC-IMS (e.g., headspace temperature and column choice) were selected, referencing prior studies or preliminary optimizations.
Results - Volatile Organic Compounds (Lines 179–217): Please elaborate on why certain compounds (e.g., (E)-2-nonenal) were more sensitive to SCC and SCCB methods. Discuss potential biochemical mechanisms behind these changes.
Please add a comparison of VOC profiles between this study and those of other similar fish species.
Results - Non-Volatile Metabolites (Lines 223–271): Please provide more interpretation of the biological significance of key metabolites (e.g., arginine, docosapentaenoic acid) and their roles in maintaining fish quality.
Discussion - Metabolic Pathways (Lines 272–356): Please discuss the potential implications of altered metabolic pathways (e.g., glycerophospholipid and amino acid metabolism) for sensory qualities such as flavor and texture.
Please highlight how these findings could influence post-harvest handling practices in the seafood industry.
Reviewer 2 Report
Comments and Suggestions for Authors
Dear Authors,
My comments are attached.
